# Disentangled Condensation for Large-scale Graphs

## Abstract

Graph condensation has emerged as an intriguing technique to save the expensive training costs of Graph Neural Networks (GNNs) by substituting a condensed small graph with the original graph. Despite the promising results achieved, previous methods usually employ an entangled paradigm of redundant parameters (nodes, edges, GNNs), which incurs complex joint optimization during condensation. This paradigm has considerably impeded the scalability of graph condensation, making it challenging to condense extremely large-scale graphs and generate high-fidelity condensed graphs. Therefore, we propose to disentangle the condensation process into a two-stage *GNN-free* paradigm, independently condensing nodes and generating edges while eliminating the need to optimize GNNs at the same time. The node condensation module avoids the complexity of GNNs by focusing on node feature alignment with anchors of the original graph, while the edge translation module constructs the edges of the condensed nodes by transferring the original structure knowledge with neighborhood anchors. This simple yet effective approach achieves at least 10 times faster than *state-of-the-art* methods with comparable accuracy on medium-scale graphs. Moreover, the proposed DisCo can successfully scale up to the Ogbn-papers100M graph containing over 100 million nodes with flexible reduction rates and improves performance on the second-largest Ogbn-products dataset by over 5%. Extensive downstream tasks and ablation study on five common datasets further demonstrate the effectiveness of the proposed DisCo framework. The source code will be made publicly available.

## CCS Concepts

• **Computing methodologies → Artificial intelligence**.

## Keywords

Graph Condensation, Large-scale Graphs, Graph Neural Networks

**ACM Reference Format:**
Anonymous Author(s). 2025. Disentangled Condensation for Large-scale Graphs. In *Proceedings of Apirl 28–May 2, 2025 (WWW '25)*. ACM, New York, NY, USA, 12 pages. https://doi.org/XXXXXXX.XXXXXXX

## 1 Introduction

Graph Neural Networks (GNNs) have emerged as a highly effective solution for modeling the non-Euclidean graph data in diverse domains such as the World Wide Web [25, 6], social networks [1, 14,

30], recommender systems [37, 27], fraud detection [3, 40, 19], molecule representations [33, 36, 38] and so on [21, 31, 47]. Despite the remarkable accomplishments of GNNs in addressing graph-related problems, they encounter substantial challenges when confronted with the ever-expanding size of real-world data. For instance, web-scale social platforms [4, 10] and e-commerce platforms [37, 46] have grown to encompass millions of nodes and billions of edges, posing prohibitively expensive costs of directly training on these large-scale graphs. The burdensome computation costs not only hinder the dedicated performance tuning of existing GNNs, but also constrain the exploration of other promising directions of GNNs like Neural Architecture Search (NAS) [8, 24, 20] and Knowledge Amalgamation [13]. As a result, there is a notable focus on reducing the computational expenses related to training GNNs on these large-scale graphs.

To address this issue, graph condensation has emerged as a promising approach to generating a compact yet informative small graph for GNNs training, significantly reducing computational costs while achieving comparable performance to training on the original graph. The *state-of-the-art* graph condensation methods mainly fall into three categories: gradient matching [7, 11, 12, 35], distribution matching [15, 16, 32], and trajectory matching [45, 41]. Gradient-matching methods aim to simulate the dynamic gradient behaviors exhibited by GNNs during the training process. In contrast, distribution-matching methods propose aligning the distribution of both graphs. Despite their potential, both of these methods face the challenge of a complex joint optimization task involving nodes, edges and GNNs, which leads to huge computation complexity. More recently, trajectory-matching methods have emerged, which assess the similarity of training trajectories of GNNs in a *graph-free* manner. However, the applicability of various GNNs on a graph-free condensed graph still poses significant computational complexity. Additionally, trajectory-based methods require multiple teacher trajectories on the original graph, which can be very demanding and burdensome, especially for large-scale graphs.

From the above analysis, we can notice that existing methods inherently adopt an entangled optimization paradigm, where redundant parameters (nodes, edges, and GNNs) are optimized simultaneously. The computation complexity of edges and GNNs causes two serious scalability problems: Firstly, obtaining a condensed graph for large-scale graphs with billions of edges is impractical, which reduces the effectiveness of graph condensation as a technique, as the most industrial scenario for graph condensation applications is graphs of this scale. Secondly, it is doubtful whether a high-fidelity condensed graph with adaptive reduction rates can be achieved. These drawbacks of the entangled strategy motivate us to develop the *GNN-free* disentangled graph condensation framework, named DisCo, which separates the generation of condensed nodes and condensed edges using the node condensation and edge translation modules, while eliminating the need to train GNNs at the same time.

The node condensation module in DisCo focuses on preserving the node feature distribution of the original graph. We achieve this

by leveraging a pre-trained node classification model on the original graph, along with class centroid alignment and anchor attachment. In the edge translation module, our goal is to preserve the topological structure of the original graph in the condensed graph. To accomplish this, we pre-train a specialized link prediction model that captures the topological structure. We then use anchors to transfer the acquired knowledge from the link prediction model to the condensed nodes, resulting in the generation of condensed edges. Notably, DisCo can successfully scale up to the Ogbn-papers100M dataset with flexible reduction rates. Extensive experiments conducted on five common datasets and Ogbn-papers100M, covering various tasks such as baseline comparison, scalability, and generalizability, further validate the effectiveness of DisCo.

In summary, our contributions are listed as follows:

- **Methodology.** We present DisCo, a novel *GNN-free* disentangled graph condensation framework that offers exceptional scalability for large-scale graphs. Notably, DisCo introduces the unique disentangled condensation strategy for the first time and effectively addresses the scalability bottleneck.
- **Scalability.** DisCo can successfully scale up to the Ogbn-papers100M dataset with flexible reduction rates, which contains over 100 million nodes and 1 billion edges. Besides, DisCo significantly improves performance on the second-largest Ogbn-products dataset by over 5%.
- **Generalizability and Time.** Experimental comparisons against baselines demonstrate that DisCo confirms the robust generalizability across five different GNN architectures, while requiring comparable or even much shorter condensation time.

## 2 Related Work

**Dataset Condensation.** Data condensation methods aim to synthesize smaller datasets from the original data while maintaining similar performance levels of models trained on them [28]. Zhao *et al.* [44, 43] propose to match the gradients with respect to model parameters and the distribution between the condensed and original datasets. Additionally, Zhao *et al.* [42] present Differentiable Siamese Augmentation, resulting in more informative synthetic images. Nguyen *et al.* [17, 18] condense datasets by utilizing Kernel Inducing Points (KIP) and approximating neural networks with kernel ridge regression (KRR). Besides, Wang *et al.* [26] propose CAFE, which enforces consistency in the statistics of features between synthetic and real samples extracted by each network layer, with the exception of the final layer.

**Graph Condensation.** Graph condensation aims to synthesize a smaller graph that effectively represents the original graph. Current graph condensation methods mainly fall into three categories: gradient matching, distribution matching, and trajectory matching. GCOND [12] achieves graph condensation by minimizing the gradient-matching loss between the gradients of training losses with respect to the GNN parameters of both the original and condensed graphs. DosCond [11], SGDD [35] and EXGC [7] are also based on gradient-matching, where DosCond simplifies the gradient-matching process, SGDD proposes to broadcast the original structure information to the condensed graph to maintain similar structure information, and EXGC focuses on convergence acceleration and pruning redundancy. GCDM [15] takes a distribution-matching approach

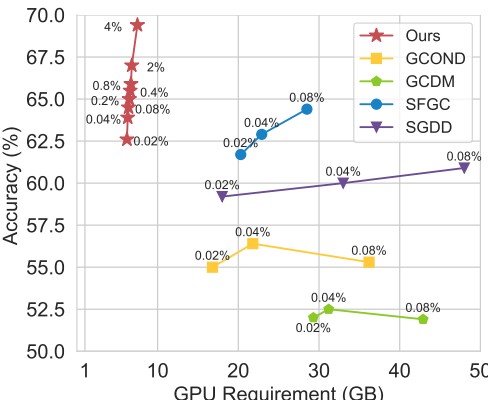

**Figure 1: The GPU requirements and accuracy of different condensation methods on the second largest Ogbn-products dataset, where "0.02%", "0.04%", "0.08%" refer to the reduction rates.**

by treating the original graph as a distribution of reception fields of specific nodes. Employing different distribution metrics from GCDM, GCEM [16] suggests aligning the distributions of node features with the eigenbasis, while SimGC [32] utilizes a pre-trained multi-layer perceptron and heuristic rules to maintain such distribution. SFGC [45] and GEOM [41] synthesize a condensed graph by matching the training trajectories of the original graph. While these methods demonstrate advantages over traditional approaches in specific scenarios, they still face significant challenges in achieving scalability for graphs of any size. One common reason is the entangled condensation strategy, which optimizes the nodes, edges, and GNNs at the same time, contributing to the substantial GPU memory requirement. Addressing this challenge would yield significant implications for various applications that involve working with large-scale graphs.

## 3 Preliminary and Pre-analysis

**Preliminary.** Suppose that there is a graph dataset $\mathcal{T} = \{A, X, Y\}$, $N$ denotes the number of nodes, $X \in \mathbb{R}^{N \times d}$ is the $d$-dimensional node feature matrix, $A \in \mathbb{R}^{N \times N}$ is the adjacency matrix, $Y \in \{0, \ldots, C - 1\}^N$ represents the node labels over $C$ classes. The target of graph condensation is to learn a condensed and informative graph dataset $\mathcal{S} = \{A', X', Y'\}$ with $A' \in \mathbb{R}^{N' \times N'}$, $X' \in \mathbb{R}^{N' \times d}$, $Y' \in \{0, \ldots, C - 1\}^{N'}$ and $N' \ll N$ so that the GNNs trained on the condensed graph are capable of achieving comparable performance to those trained on the original graph. Thus, the objective function can be formulated as follows:

$$\min_{\mathcal{S}} \mathcal{L}\left(\text{GNN}_{\theta_{\mathcal{S}}}(A, X), Y\right),$$

$$\text{s.t} \quad \theta_{\mathcal{S}} = \arg\min_{\theta} \mathcal{L}(\text{GNN}_{\theta}(A', X'), Y'), \quad (1)$$

where $\text{GNN}_{\theta}$ represents the GNN model parameterized with $\theta$, $\theta_{\mathcal{S}}$ denotes the parameters of the model trained on $\mathcal{S}$, $\mathcal{L}$ denotes the loss function such as cross-entropy loss. Directly optimizing the

objective function in this bi-level optimization problem is challenging. Additionally, it is also expensive to compute the second-order derivatives with respect to GNN parameters.

**Pre-analysis.** Since the triple parameters of the condensed graph are closely entangled together, existing methods resort to two kinds of methods to simplify the condensation process. $Y'$ are all predefined according to the class distribution. $X'$ is usually initialized as nodes from the original graph sampled randomly or by K-Center algorithm [23]. On the one hand, gradient- [7, 11, 12, 35] and distribution-matching [15] methods parameterize the adjacency matrix $A'$ as a function of $X'$ using a pairwise multi-layer perceptron (MLP), written as

$$A'_{ij} = \text{Sigmoid}(\frac{\text{MLP}_\phi([X'_i; X'_j]) + \text{MLP}_\phi([X'_j; X'_i])}{2}), \quad (2)$$

where $[X'_i; X'_j]$ denotes the concatenation of the $i^{\text{th}}$ and $j^{\text{th}}$ nodes features, $A'_{ij}$ denotes the edge weight of the $i^{\text{th}}$ and $j^{\text{th}}$ nodes. Nevertheless, the convolution operation of GNNs involves both nodes and edges within the condensed graph, necessitating the simultaneous optimization of all these parameters during condensation, which inherently leads to a complex and challenging joint optimization problem. On the other hand, trajectory-matching methods [45] simplify the adjacency matrix $A'$ as an identity matrix and align the training trajectories between GNNs of the original graph and the condensed graph. However, imposing this *graph-free* constraint on $A'$ seems contradictory to the fundamental design principles of GNNs. Additionally, these methods still require optimizing nodes and GNNs simultaneously. As depicted in Figure 1, the entanglement of various parameters results in significant GPU memory consumption, limiting their scalability when applied to large-scale graphs. In contrast, our method experiences only a slight increase in GPU memory usage even when the reduction rate is increased by a hundredfold.

## 4 METHOD
### 4.1 Overall Framework

The drawbacks of the entangled optimization paradigm motivate us to devise a *GNN-free* disentangled condensation framework, DisCo. Our framework contains two complementary modules: the node condensation module and the edge translation module, which independently focus on condensing nodes and generating edges respectively, while eliminating the need to train GNNs at the same time, thus boosting the scalability of condensation methods. The primary goal of the node condensation module is to preserve the node feature distribution of the original graph. Thus, we employ a pre-trained node classification MLP along with class centroid alignment and anchor attachment to ensure the preservation of the feature distribution in the condensed nodes. In the edge translation module, our objective is to preserve the topological structure of the original graph in the condensed graph. Consequently, we pre-train a specialized link prediction model that captures the intricate relationships between nodes within the original graph structure. We then transfer this knowledge to the condensed nodes, enabling the efficient generation of condensed edges that accurately reflect the original graph's topology. The overall pipeline of the proposed DisCo framework is shown in Figure 2.

### 4.2 Node Condensation

The node condensation module is designed to generate a condensed node set that accurately represents the original nodes for training GNNs. The success of GNNs heavily relies on the distribution of node features, so the primary goal of the node condensation module is to preserve the original node feature distribution. But how can we preserve such distribution without including external parameters (like complex GNNs) at the same time, ensuring lightweight node condensation? To tackle this challenge, we introduce a novel node condensation methodology that employs a pre-trained MLP with two regularization terms to harmonize the node feature distributions across both graphs.

**MLP Alignment.** Firstly, we pre-train a node classification MLP model on the original nodes by optimizing the corresponding classification loss:

$$L_{\text{ori}} = \mathcal{L}(\text{MLP}_\phi(X), Y), \quad (3)$$

where $\text{MLP}_\phi$ denotes the MLP classification model and $\mathcal{L}$ represents the classification loss. Subsequently, the initial condensed nodes are sampled using K-Center from the original graph. Then we leverage the pre-trained MLP to optimize the condensed node features by minimizing the classification loss, thereby ensuring that the feature spaces of both graphs are effectively aligned. The corresponding can be formulated as follows:

$$L_{\text{cls}} = \mathcal{L}(\text{MLP}_\phi(X'), Y'), \quad (4)$$

where $\mathcal{L}$ represents the classification loss.

**Class Centroid Alignment.** The first regularization term is class centroid alignment. Class centroid refers to the average node features of each class, which is a crucial indicator of the distribution of node features. It is essential to ensure that the condensed nodes have comparable class centroids in order to preserve the same feature distribution, so the first regularizer term is:

$$L_{\text{alg}} = \sum_{c=0}^{C-1} \text{MSE}(\mu_c, \mu'_c), \quad (5)$$

where $\mu_c$ and $\mu'_c$ denote the centroids of $c$ class of the original and condensed nodes respectively.

**Anchor Attachment.** The second regularization term is anchor attachment. We want each condensed node to be adaptively mapped to one or a few nodes in the original graph, thus achieving more accurate feature preservation and explainable node condensation. To achieve this, we introduce a set of important nodes named anchors, which refer to the $k$ nearest original nodes that belong to the same class as each condensed node. During the condensation process, we enable each condensed node to be adaptively attached to its $k$-nearest anchors among the original nodes using a distance term:

$$L_{\text{anc}} = \sum_{i=1}^{N'} \sum_{j=1}^{M} D(X'_i, Z_{ij}), \quad (6)$$

where $Z_{ij}$ is the $j^{\text{th}}$ anchors of $X'_i$, $M$ is the anchor number, and $D$ is the distance function like L2 norm. By minimizing the distance during the node condensation, we encourage each condensed node to be closely associated with its most relevant anchor nodes from the original graph. This helps to ensure better feature preservation and condensation interpretability.

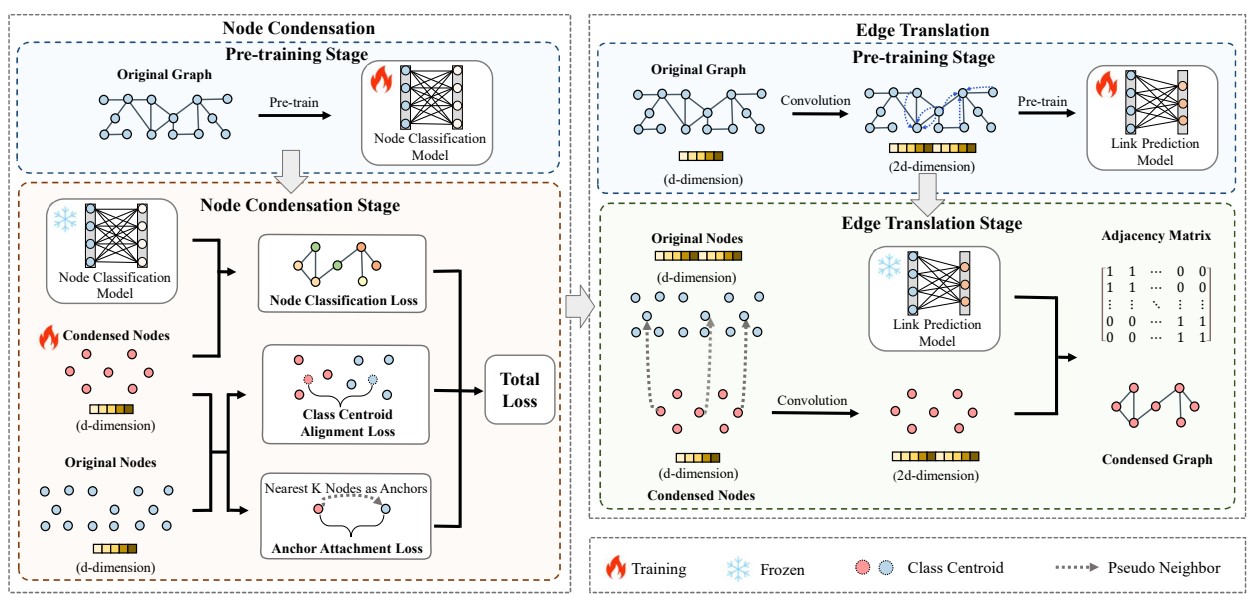

**Figure 2: An illustrative diagram of the proposed DisCo framework.**

With the above regularization terms, the final node condensation objective function is written as:

$$\underset{X'}{\arg\min}(L_{\text{cls}} + \alpha L_{\text{alg}} + \beta L_{\text{anc}}), \quad (7)$$

where $\alpha$ and $\beta$ are hyperparameters that control the weights of two regularizers. More theoretical analysis about node condensation can be found in Appendix D.1.

### 4.3 Edge Translation

Following node condensation, the subsequent step is to translate the topology of the original graph into the condensed graph. As previously indicated, existing methods parameterize condensed edges as a function of condensed nodes and simultaneously optimize nodes, edges, and GNNs, which is highly unscalable due to the immense computational complexity involved. So we introduce a novel edge translation method that eliminates the need for optimizing condensed edges. The edge translation process can be divided into two stages: (1) preserving the topology of the original graph by pre-training a specialized link prediction model considering neighbor information; (2) transferring the link prediction model to the condensed nodes with the aid of pseudo neighbors, obtaining the condensed edges.

**Pre-training.** In the first step, our aim is to preserve the topology of the original graph through a link prediction model. However, the absence of predefined edges in the condensed nodes makes it difficult to utilize a GNN-based link prediction model. Existing condensation methods mostly rely on a simple MLP like Eq. (2) without considering neighbor information to predict the edges of the condensed graph, which may not fully capture the relationships between nodes. To investigate this critical issue, we conduct an experiment on the original Ogbn-products and Reddit datasets to compare the effectiveness of link prediction models with and without neighbor information. Based on Table 1, it becomes evident that disregarding

**Table 1: The performance comparison between link prediction models with and without neighbor information. "With" refers to our specialized model.**

|  | Ogbn-products | | | Reddit | | |
|---|---|---|---|---|---|---|
|  | Accuracy | Precision | Recall | Accuracy | Precision | Recall |
| Without | 0.892 | 0.837 | 0.705 | 0.925 | 0.864 | 0.831 |
| With | 0.981 | 0.962 | 0.963 | 0.965 | 0.926 | 0.935 |

neighbor information can significantly undermine the effectiveness of the link prediction model, thereby compromising the preservation of the topology. Therefore, we propose a specialized link prediction model to incorporate neighbor information from the original graph. We first perform aggregation to attract neighbor information using an aggregator (e.g. mean, sum and max aggregator) and obtain the neighbor feature $h_{N(v)}$ of each node in the original graph. Then we concatenate the node feature with the above neighbor feature and obtain the convolved node feature: $H_v = [h_v; h_{N(v)}]$. After that, we can predict the link existence $e_{ij}$ as a function of $H$, denoted as $g_\vartheta$ using the following:

$$e_{ij} = \text{Sigmoid}\left(\frac{\text{MLP}_\vartheta([H_i; H_j]) + \text{MLP}_\vartheta([H_j; H_i])}{2}\right), \quad (8)$$

where $\text{MLP}_\vartheta$ represents the Multilayer Perceptron, $[H_i; H_j]$ denotes the concatenation of the convolved node features of the $i$th and $j$th original nodes, and $e_{ij}$ represents the edge exsitence between the $i$th and $j$th original nodes. We can pre-train the link prediction model using Eq. (8) and a binary cross-entropy (BCE) loss function.

**Translation.** To transfer the pre-trained link prediction model to the condensed nodes, our strategy involves identifying suitable pseudo neighbors for them. As mentioned in the node condensation section, each condensed node has $k$-nearest anchors in the original nodes.

**Table 2: The performance comparison of our proposed DisCo and baselines on different datasets under various reduction rates. Performance is reported as test accuracy (%). ± corresponds to one standard deviation of the average evaluation over 5 trials. "Whole Dataset" refers to training with the whole dataset without graph condensation.**

| Dataset | Reduction Rate | Baselines | | | | | | | Proposed | Whole Dataset |
|---|---|---|---|---|---|---|---|---|---|---|
| | | Random | Herding | K-Center | GCOND | GCDM | SFGC | SGDD | DisCo | |
| Cora | 1.3% | 63.6 ± 3.7 | 67.0 ± 1.3 | 64.0 ± 2.3 | 79.8 ± 1.3 | 69.4 ± 1.3 | **80.1 ±0.7** | **80.1 ± 0.4** | 76.9 ± 0.8 | |
| | 2.6% | 72.8 ± 1.1 | 73.4 ± 1.0 | 73.2 ± 1.2 | 80.1 ± 0.6 | 77.2 ± 0.4 | 80.6±0.8 | **81.7 ± 0.5** | 78.7 ± 0.3 | 82.5 ± 1.2 |
| | 5.2% | 76.8 ± 0.1 | 76.8 ± 0.1 | 76.7 ± 0.1 | 79.3 ± 0.3 | 79.4 ± 0.1 | 80.4±1.6 | **81.6 ± 0.8** | 78.8 ± 0.5 | |
| Ogbn-arxiv | 0.05% | 47.1 ± 3.9 | 52.4 ± 1.8 | 47.2 ± 3.0 | 59.2 ± 1.1 | 53.1 ± 2.9 | **65.5 ± 0.7** | 60.8±1.3 | 64.0 ± 0.7 | |
| | 0.25% | 57.3 ± 1.1 | 58.6 ± 1.2 | 56.8 ± 0.8 | 63.2 ± 0.3 | 59.6 ± 0.4 | 66.1 ± 0.4 | **66.3±0.7** | 65.9 ± 0.5 | 71.1 ± 0.2 |
| | 0.5% | 60.0 ± 0.9 | 60.4 ± 0.8 | 60.3 ± 0.4 | 64.0 ± 0.4 | 62.4 ± 0.1 | **66.8 ± 0.4** | 66.3±0.7 | 66.2 ± 0.1 | |
| Ogbn-products | 0.02% | 53.5 ± 1.3 | 55.1 ± 0.3 | 48.5 ± 0.2 | 55.0 ± 0.8 | 53.0 ± 1.9 | 61.7 ± 0.5 | 57.2±2.0 | **62.2 ± 0.5** | |
| | 0.04% | 58.5 ± 0.7 | 59.1 ± 0.1 | 53.3 ± 0.4 | 56.4 ± 1.0 | 53.5 ± 1.1 | 62.9 ± 1.2 | 58.1±1.9 | **64.5 ± 0.7** | 74.0 ± 0.1 |
| | 0.08% | 63.0 ± 1.2 | 53.6 ± 0.7 | 62.4 ± 0.5 | 55.3 ± 0.3 | 52.9 ± 0.9 | 64.4 ± 0.4 | 59.3±1.7 | **64.6 ± 0.8** | |
| Reddit | 0.05% | 46.1 ± 4.4 | 53.1 ± 2.5 | 46.6 ± 2.3 | 88.0 ± 1.8 | 73.9 ± 2.0 | 89.7 ± 0.2 | 90.5±2.1 | **91.4 ± 0.2** | |
| | 0.1% | 58.0 ± 2.2 | 62.7 ± 1.0 | 53.0 ± 3.3 | 89.6 ± 0.7 | 76.4 ± 2.8 | 90.0 ± 0.3 | **91.8±1.9** | **91.8 ± 0.3** | 93.9 ± 0.1 |
| | 0.2% | 66.3 ± 1.9 | 71.0 ± 1.6 | 58.5 ± 2.1 | 90.1 ± 0.5 | 81.9 ± 1.6 | 90.3 ± 0.3 | 91.6±1.8 | **91.7 ± 0.3** | |
| Reddit2 | 0.05% | 48.3 ± 6.4 | 46.9 ± 1.2 | 43.2 ± 3.2 | 79.1 ± 2.2 | 73.5 ± 4.7 | 84.4 ± 1.7 | 86.7±0.8 | **90.9 ± 0.4** | |
| | 0.1% | 57.8 ± 3.1 | 62.5 ± 2.8 | 51.9 ± 0.7 | 82.4 ± 1.0 | 75.4 ± 1.8 | 88.1 ± 1.9 | 85.8±1.1 | **90.8 ± 0.5** | 93.5 ± 0.1 |
| | 0.2% | 65.5 ± 2.5 | 71.4 ± 1.6 | 57.4 ± 1.8 | 80.6 ± 0.4 | 80.8 ± 3.1 | 88.6 ± 1.1 | 85.4±0.6 | **91.3 ± 0.3** | |

We can utilize these anchors as pseudo neighbors for the condensed nodes. By performing the same aggregation on the anchor nodes in the original graph, we can obtain the neighbor feature $h'_{N(v)}$ and the convolved node feature $H'_v = [h'_v; h'_{N(v)}]$ of each condensed node. Finally, we can apply the pre-trained link prediction model to the $H'_v$ and obtain the condensed edges using $g'_\vartheta$:

$$a'_{ij} = \text{Sigmoid}(\frac{\text{MLP}_\vartheta([H'_i; H'_j]) + \text{MLP}_\vartheta([H'_j; H'_i])}{2}), \quad (9)$$

$$A'_{ij} = \begin{cases} a'_{ij}, & \text{if } a'_{ij} \geq \delta, \\ 0, & \text{if } a'_{ij} < \delta, \end{cases} \quad (10)$$

where $\delta$ is the threshold that decides whether there is an edge, $A'_{ij}$ denotes the final edge weight of the $i^{\text{th}}$ and $j^{\text{th}}$ condensed nodes. Finally, we can obtain the condensed edges and form the final condensed graph. Although the number of edges may have changed, these edges still capture the underlying topological structure of the original graph. More theoretical analysis about edge translation can be found in Appendix D.2.

## 5 Experiments

### 5.1 Experiment Settings

**Datasets.** We evaluate the downstream GNN performance of the condensed graphs on six datasets, four of which are transductive datasets such as Cora [14], Ogbn-arxiv, Ogbn-products and Ogbn-papers100M [10] and two of which are inductive datasets such as Reddit [9] and Reddit2 [39].

**Baselines.** We compare our approach with several baseline methods: (1) three coreset methods including Random, Herding [29] and K-Center [23]; (2) four graph condensation methods including GCOND [12], GCDM [15], SFGC [45] and SGDD [35].

**Implementation.** The experiment process is divided into three stages: (1) obtaining the condensed graph with a reduction rate $r$, (2) training a $\text{GNN}_S$ with the condensed graph, then selecting the best-performed model using the original validation set, and (3) evaluating the model with the original test set. The experiments for Cora, Ogbn-arxiv, Reddit and Reddit2 are performed using a single Quadro P6000, experiments for the others are performed using a single NVIDIA A40 due to the substantial GPU memory demand.

### 5.2 Prediction Accuracy

To evaluate the downstream performance of the condensed graphs generated by different condensation methods, we report the test accuracies for each method on different datasets with different reduction rates in Table 2. Based on the results, the following observations can be made:

▶ Observation 1. DisCo has not only shown comparable performance but even superior performance on larger-scale graphs when compared to the current *state-of-the-art* methods. This observation suggests that the existing methods are not scalable and may perform poorly on large-scale graphs. In contrast, the disentangled Disco consistently achieves the best performance as the graph scale increases.

▶ Observation 2. DisCo achieves exceptional performance even with significantly low reduction rates. Moreover, as the reduction rate increases, the performance of DisCo improves and gradually approaches that of the entire dataset. This indicates the importance of the disentangled paradigm, as it allows for flexible reduction rates of the original graph.

▶ Observation 3. Disco's performance on Cora is subpar. The reason is that the node condensation module requires labeled nodes to ensure the node distribution's integrity. However, labeled nodes are fewer than one-tenth of the total on Cora, which poses challenges

in preserving the full node distribution. It is important to note that small datasets do not require graph condensation and it's more vital to highlight our SOTA performance on large-scale graphs.

## 5.3 Condensation Time

In this section, we compare the condensation time of our method with four condensation methods. To provide a comprehensive analysis, we report the time taken for node condensation and edge translation of our method separately, and then present the total condensation time. It can be observed from Table 3 that the node condensation process is highly efficient, with an extremely short time. While the edge translation process takes longer than node condensation, the time spent is largely devoted to pre-training a simple MLP-based link prediction model, which remains efficient and stable even in large-scale graphs like Ogbn-products. Upon examining the total time, DisCo demonstrates a significantly shorter time compared to existing graph condensation methods in the Ogbn-arxiv and Ogbn-products datasets. Notably, the condensation time for Ogbn-products is reduced to just one-fourth of the time required by the second most efficient method, demonstrating its efficiency benefits on large-scale graphs. Analyzing the average test accuracy rank and average condensation time rank, we notice that while SFGC and SGDD achieve comparable accuracy ranks to DisCo, they require significantly longer condensation time. In contrast, although GCDM exhibits faster training speed than DisCo on the dense graph Reddit, it consistently delivers the poorest performance across all datasets. Notably, our method successfully strikes a balance between condensation accuracy and condensation time, achieving high performance on both fronts. The time complexity analysis presented in Appendix B.3 substantiates the efficiency of our method, demonstrating why it operates in a notably shorter duration.

It is important to emphasize a significant advantage of DisCo lies in its ability to pre-train the link prediction model only once and subsequently apply it to different condensed nodes. This makes DisCo a far more efficient option, particularly in scenarios where graph condensation needs to be performed multiple times with varying reduction rates.

## 5.4 Scalability

In this section, we will investigate the scalability of DisCo through two experiments, evaluating its ability to condense extremely large-scale graphs and generate high-fidelity condensed graphs. First, we assess DisCo's ability to condense extremely large-scale graphs. To this end, we utilize the Ogbn-papers100M dataset, which comprises over 100 million nodes and 1 billion edges. Since existing graph condensation methods are incapable of handling such large-scale graphs due to memory constraints, we compare DisCo with coreset methods, including Random, Herding, and K-Center, using a Simplified Graph Convolution (SGC) model. The results presented in Table 4 demonstrate the effectiveness of DisCo in condensing the Ogbn-papers100M dataset at various reduction rates while maintaining a significant level of performance. Notably, DisCo achieves performance improvements of over 7% at all reduction rates compared to coreset methods, showcasing its superior effectiveness in condensing super large-scale graphs.

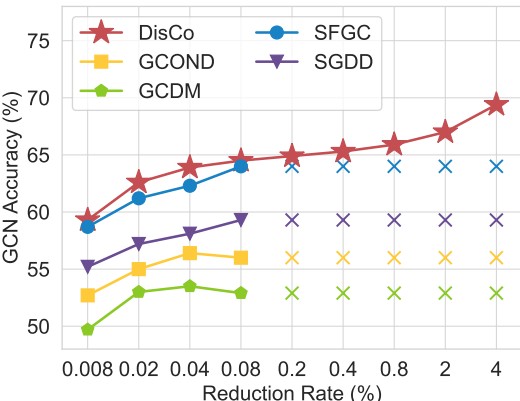

**Figure 3: The performance comparison of DisCo and baselines using GCN under different reduction rates on Ogbn-products. "X" stands for out of GPU memory (49GB).**

Scalability in the graph condensation problem also involves the capability to obtain high-fidelity condensed graphs. Therefore, in the second experiment, we compare the effectiveness of different condensation methods in condensing high-fidelity graphs by evaluating their performance at various reduction rates using Ogbn-products. The results presented in Figure 3 demonstrate that DisCo consistently outperforms other condensation methods across all reduction rates on Ogbn-products. Additionally, DisCo exhibits a much higher upper limit for the reduction rate, exceeding 4%, allowing for the generation of high-fidelity condensed graphs. In contrast, the other methods have upper limits of only 0.08%. Beyond a reduction rate of 0.08%, these methods cannot proceed due to insufficient GPU resources, resulting in low-fidelity condensed graphs. Furthermore, the optimal GCN performance achieved by DisCo is 69.4%, which is over 5% higher than the optimal performance of the other methods. This indicates that a high-fidelity condensed graph is crucial in achieving excellent GNN prediction accuracy, making scalability an even more meaningful topic to consider. These findings highlight the strong ability of DisCo to condense high-fidelity graphs compared to other methods.

## 5.5 Generalizability

In this section, we illustrate the generalizability of different condensation methods using different GNN models (GCN, SGC, GraphSAGE [9], GIN [33], JKNet [34]). The results presented in Table 5 demonstrate the generalization capabilities of DisCo across various GNN architectures and datasets. The condensed graph generated by DisCo exhibits performance superiority over other methods across the majority of datasets and models, except for the Cora dataset. The small size of the Cora dataset poses challenges in maintaining the node distribution during node condensation in our method. Importantly, DisCo demonstrates consistently higher average performance compared to other methods across almost all datasets. Furthermore, MLP consistently demonstrates the lowest performance among all the models across all datasets and condensation methods. This highlights the crucial role of edges in the condensed graph and emphasizes their indispensability.

**Table 3: The condensation time (seconds) of our proposed DisCo and baselines on four large-scale datasets (#Node/#Edge). "Acc Rank" means the average test accuracy rank among all baselines, while "Time Rank" means the average condensation time rank.**

| | | Ogbn-arxiv (169K/1M) | | | Ogbn-products (2M/61M) | | | Reddit (232K/114M) | | | Reddit2 (232K/23M) | | | Acc | Time |
|---|---|---|---|---|---|---|---|---|---|---|---|---|---|---|---|
| | | 0.05% | 0.25% | 0.5% | 0.02% | 0.04% | 0.08% | 0.05% | 0.1% | 0.2% | 0.05% | 0.1% | 0.2% | Rank | Rank |
| GCOND | Total | 13,224 | 14,292 | 18,885 | 20,092 | 25,444 | 25,818 | 15,816 | 16,320 | 19,225 | 10,228 | 10,338 | 11,138 | 3.9 | 3.0 |
| GCDM | Total | 1,544 | 5,413 | 13,602 | 11,022 | 12,316 | 13,292 | 1,912 | 2,372 | 4,670 | 1,615 | 1,833 | 4,574 | 4.7 | 1.7 |
| SFGC | Total | 64,260 | 67,873 | 70,128 | 128,904 | 130,739 | 132,606 | 159,206 | 160,190 | 161,044 | 124,774 | 125,331 | 126,071 | **2.0** | 5.0 |
| SGDD | Total | 15,703 | 17,915 | 21,736 | 28,528 | 39,579 | 59,622 | 46,096 | 54,165 | 55,274 | 35,304 | 38,233 | 40,987 | 2.1 | 4.0 |
| DisCo | Node | 99 | 106 | 107 | 216 | 251 | 351 | 315 | 326 | 368 | 327 | 356 | 382 | | |
| | Edge | 1,244 | 1,244 | 1,244 | 2,760 | 2,760 | 2,760 | 2,798 | 2,798 | 2,798 | 1,654 | 1,654 | 1,654 | **2.0** | **1.3** |
| | Total | 1,343 | 1,350 | 1,351 | 2,976 | 3,011 | 3,111 | 3,113 | 3,124 | 3,166 | 1,981 | 2,010 | 2,136 | | |

**Table 4: The performance comparison of DisCo and coreset methods using SGC on Ogbn-papers100M. "Whole Dataset" refers to training with the whole dataset.**

| | Ogbn-papers100M | | | | |
|---|---|---|---|---|---|
| | 0.005% | 0.01% | 0.02% | 0.05% | Whole Dataset |
| Random | 12.8 | 17.8 | 29.8 | 37.5 | |
| Herding | 21.3 | 26.8 | 36.2 | 43.7 | 63.3 |
| K-Center | 8.7 | 10.4 | 17.2 | 26.3 | |
| DisCo | **48.3** | **48.7** | **49.6** | **50.9** | |

**Table 5: The generalization capabilities of different condensation methods using a high reduction rate. "SAGE" stands for GraphSAGE, and "Avg." stands for the average test accuracy of five GNNs.**

| | Methods | MLP | GCN | SGC | SAGE | GIN | JKNet | Avg. |
|---|---|---|---|---|---|---|---|---|
| Cora (2.6%) | GCOND | 73.1 | 80.1 | **79.3** | 78.2 | 66.5 | **80.7** | 77.0 |
| | GCDM | 69.7 | 77.2 | 75.0 | 73.4 | 63.9 | 77.8 | 73.5 |
| | SFGC | **81.1** | **81.1** | 79.1 | **81.9** | 72.9 | 79.9 | **79.0** |
| | SGDD | 76.8 | 79.8 | 78.5 | 80.4 | 72.8 | 76.9 | 77.7 |
| | DisCo | 59.5 | 78.6 | 75.0 | 75.6 | **74.2** | 78.7 | 76.4 |
| Ogbn-arxiv (0.5%) | GCOND | 43.8 | 64.0 | 63.6 | 55.9 | 60.1 | 61.6 | 61.0 |
| | GCDM | 41.8 | 61.7 | 60.1 | 53.0 | 58.4 | 57.2 | 58.1 |
| | SFGC | 46.6 | **67.6** | 63.8 | 63.8 | 61.9 | 65.7 | 64.6 |
| | SGDD | 36.9 | 65.6 | 62.2 | 53.9 | 59.1 | 60.1 | 60.2 |
| | DisCo | **49.5** | 66.2 | **64.9** | **64.2** | **63.2** | **66.2** | **64.9** |
| Ogbn-products (0.04%) | GCOND | 33.9 | 56.4 | 52.3 | 44.5 | 50.5 | 46.3 | 50.0 |
| | GCDM | 37.1 | 54.4 | 49.0 | 48.1 | 50.4 | 49.3 | 50.2 |
| | SFGC | 40.9 | 64.2 | 60.4 | **60.4** | 58.9 | 61.6 | 58.3 |
| | SGDD | 25.5 | 57.0 | 50.1 | 51.5 | 51.3 | 49.5 | 51.9 |
| | DisCo | **45.3** | **65.1** | 59.1 | 60.2 | **63.2** | 61.0 | **61.9** |
| Reddit (0.2%) | GCOND | 48.4 | 91.7 | 92.2 | 73.0 | 83.6 | 87.3 | 85.6 |
| | GCDM | 40.5 | 83.3 | 79.9 | 55.0 | 78.8 | 77.3 | 74.9 |
| | SFGC | 45.4 | 88.7 | 87.6 | 84.5 | 80.3 | 88.2 | 85.7 |
| | SGDD | 24.8 | 89.8 | 87.5 | 73.7 | 85.2 | 88.9 | 85.0 |
| | DisCo | **55.2** | **91.8** | **92.5** | **89.0** | **85.4** | **90.5** | **89.8** |
| Reddit2 (0.2%) | GCOND | 35.7 | 82.4 | 77.1 | 59.8 | 79.6 | 73.0 | 74.4 |
| | GCDM | 32.5 | 84.7 | 78.0 | 55.3 | 75.3 | 70.6 | 72.8 |
| | SFGC | 42.6 | 88.0 | 86.8 | 77.9 | 74.8 | 86.0 | 82.7 |
| | SGDD | 25.1 | 86.0 | 87.5 | 73.1 | 79.4 | 84.7 | 82.1 |
| | DisCo | **50.2** | **91.6** | **92.0** | **88.2** | **87.0** | **89.5** | **89.7** |

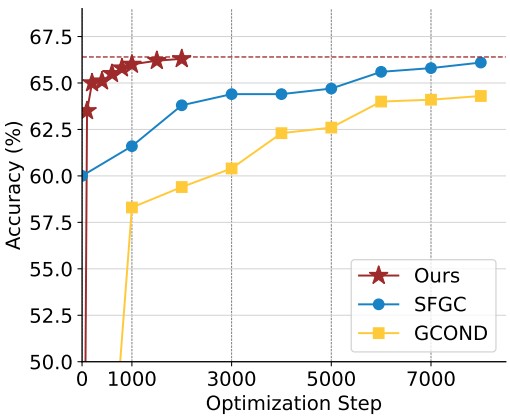

**Figure 4: The optimization step and performance comparison of DisCo and baselines on Ogbn-arxiv (0.5%) using GCN.**

## 5.6 Optimization Step

We compare various methods' optimization steps and corresponding GCN test accuracies on the Ogbn-arxiv (0.5%) dataset. As shown in Table. 4, DisCo requires significantly fewer optimization steps to achieve optimal performance. This efficiency stems from our method's use of first-order optimization, in which we directly optimize the targeted condensed graph without considering other parameters. In contrast, second-order optimization methods require training an inner loop GNN to obtain a trajectory or gradient. These GNN metrics are then used to proceed with alignment, meaning that the condensed graph heavily relies on the trained GNNs and their different initializations. This approach results in substantial computational demands and instability. Notably, the solution achieved through first-order optimization demonstrates comparable or even superior performance to other methods. This makes DisCo an easily optimized and high-performance choice for graph condensation.

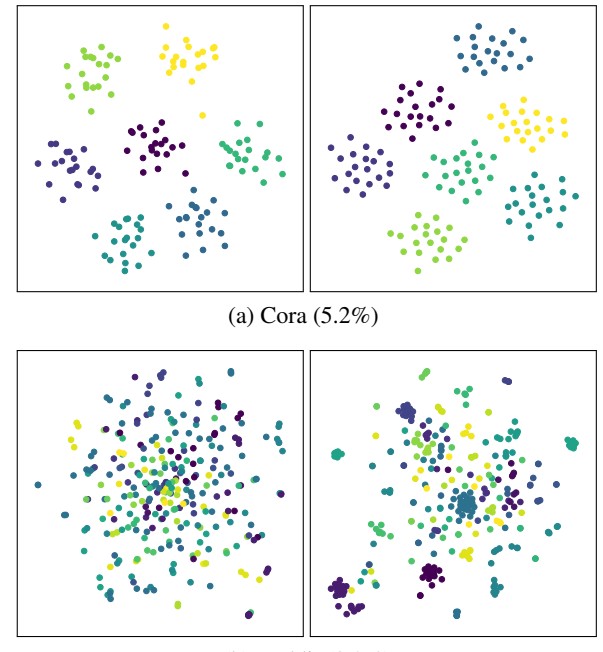

(a) Cora (5.2%)

(b) Reddit (0.2%)

**Figure 5: The T-SNE plots of the condensed nodes. Left: SFGC(NeurIPS 2023) [45], Right: DisCo.**

## 5.7 Visualization and Analysis

We present visualization results for the Cora (5.2%) and Reddit (0.2%) datasets, comparing three clustering indexes and homophily with the SOTA method SFGC and GCOND. The indexes we use here include Silhouette Coefficient [22], Davies-Bouldin [5] and Calinski-Harabasz Index [2]. Figure 5 and Table 6 illustrate that the condensed graphs of Cora displays clear clustering tendencies under both SFGC and DisCo. Notably, DisCo's condensed graph for Reddit demonstrates significantly enhanced clustering patterns, highlighting that our method's advantage in maintaining the node distribution in large-scale graphs. Moreover, DisCo surpasses GCOND in preserving the homophily of the original graph, consistently upholding a high level of homophily similarity. This indicates that the condensed graphs exhibit similar connectivity patterns and topology to the original graphs. These findings collectively provide robust evidence that our disentangled approach can effectively capture high-quality node features and topology, delivering superior downstream performance.

## 5.8 Ablation Study

In this subsection, we aim to validate the effectiveness of the link prediction model utilized in DisCo by comparing it with a traditional method Eq. (2) that ignores neighbor information (referred to as simple in Figure 6). Based on the obtained results in Figure 6, we can observe that our link prediction method consistently outperforms the simple link prediction method across all datasets and reduction rates. Furthermore, in the Ogbn-products dataset, the performance gap between the two methods increases as the reduction rate increases, with the gap exceeding 2% when the reduction rate reaches 0.8%. These findings highlight the effectiveness of our link prediction

**Table 6: Comparison of clustering patterns and homophily between our method and GCOND, SFGC. SC (Silhouette Coefficient), DB (Davies-Bouldin Index), and CH (Calinski-Harabasz Index) are used as clustering metrics. ↑ indicates that higher values represent better clustering patterns, while ↓ indicates the opposite.**

|  | SC (↑) | | DB (↓) | | CH (↑) | | Homophily | | |
|--|--------|--|--------|--|--------|--|-----------|--|--|
|  | SFGC | DisCo | SFGC | DisCo | SFGC | DisCo | GCOND | DisCo | Whole |
| Cora | **0.09** | **0.09** | **2.56** | 2.76 | **6.27** | 6.03 | 0.79 | 0.97 | 0.81 |
| Reddit | -0.32 | **-0.16** | **4.68** | 5.17 | 2.41 | **3.22** | 0.04 | 0.88 | 0.78 |

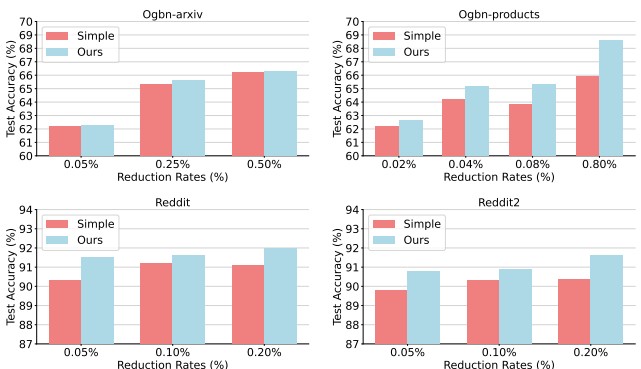

**Figure 6: Evaluation of the proposed link prediction model. "Simple" refers to simple link prediction model.**

model in preserving the topological structure, primarily due to its ability to capture neighbor information.

## 6 Conclusion

This paper introduces DisCo, a novel *GNN-free* disentangled graph condensation framework designed to address the entangled optimization issues observed in existing methods, which greatly limit their scalability. DisCo obtains the condensed nodes and edges via the node condensation and edge translation modules respectively, while eliminating the need to train GNNs at the same time. This simple yet remarkably effective approach results in significant GPU savings in terms of condensation. Consequently, DisCo scales successfully to Ogbn-papers100M, a dataset consisting of over 100 million nodes and 1 billion edges, providing highly tunable reduction rates. Besides, DisCo significantly improves performance on the second-largest Ogbn-products dataset by over 5%. Extensive experiments conducted on five datasets demonstrate that DisCo consistently performs on par with or outperforms state-of-the-art baselines while maintaining an extremely low cost.

**Limitations and Future Work.** Although disentangling the condensation process can offer scalability advantages, this disentangled paradigm lacks explicit theoretical robustness and the intuitive logic found in gradient or trajectory matching methods. In future work, we will focus on enhancing the theoretical robustness of this framework.

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

# Appendix

In this appendix, we provide more details of the proposed DisCo in terms of detailed algorithm, dataset statistics, baselines and experimental settings with some additional analysis and experiments.

## A  Algorithm

The whole condensation process is divided into two parts in order: node condensation and edge translation. We show the detailed algorithm of DisCo in Algorithm 1.

---

**Algorithm 1:** The Proposed DisCo Framework

**Input:** Original graph $\mathcal{T} = (A, X, Y)$.
Initialize $X'$ randomly and pre-define condensed labels $Y'$.
**Node Condensation:**
Pre-train a node classification model $\text{MLP}_\phi$.
**for** $t = 0, \ldots, T - 1$ **do**
    $L_{\text{cls}} = \mathcal{L}(\text{MLP}_\phi(X'), Y')$.
    $L_{\text{alg}} = 0$.
    **for** $c \in \{0, \ldots, C - 1\}$ **do**
        Compute the proportion of this class $\lambda_c$.
        Compute the feature mean $\mu$ and $\mu'$ of $c$ class.
        $L_{\text{alignment}} += \lambda_c * \text{MSE}(\mu_c, \mu'_c)$
    $L_{\text{anc}} = \sum_{i=1}^{N'} \sum_{j=1}^{M} D(X'_i, Z_{ij})$.
    $L = L_{\text{cls}} + \alpha L_{\text{alg}} + \beta L_{\text{anc}}$.
    Update $X' = X' - \eta \nabla_{X'} L$.
**Edge Translation:**
Obtain convolved original node features $H_v$.
Pre-train a link prediction model $g_\vartheta$ with Eq. (8).
Obtain convolved condensed node features $H'_v$.
Predict the condensed edges with Eq. (9) and (10).
**Output:** $\mathcal{S} = (A', X', Y')$.

---

## B  Datasets, Baselines and Hyperparameters

### B.1  Datasets

We evaluate the condensation performance of our method on six datasets, four of which are transductive datasets such as Cora [14], Ogbn-arxiv, Ogbn-products and Ogbn-papers100M [10] and two of which are inductive datasets like Reddit [9] and Reddit2 [39]. The Cora dataset can be accessed using the pytorch-geometric library. All the Ogbn datasets can be accessed using the Ogb library. And the inductive datasets can also be accessed with the pytorch-geometric library. We directly use public splits to split them into the training, validation, and test sets. The detailed statistics are summarized in Table 7. In the condensation process, We make full use of the complete original graph and training set labels for transductive datasets, whereas we use only the training set subgraph and corresponding labels for inductive datasets.

### B.2  Baselines

We compare our DisCo framework with several baseline methods: (1) three coreset methods including Random, Herding [29] and K-Center [23]; (2) four graph condensation methods including GCOND [12], GCDM [15], SFGC [45] and SGDD [35]. For these coreset methods, we derive the condensed graph from the original graph by identifying a small set of core nodes using the three coreset methods and then inducing a subgraph from these selected nodes. As for the condensation methods, since the source code for GCDM cannot be provided by the authors, we have to re-implement our own version of GCDM from scratch. The experimental results demonstrate that our implemented GCDM yields consistent results with those reported in the original paper [15]. To ensure reliability, we employ the published results of [15].

### B.3  Hyperparameter Settings

In the baseline comparison experiment, we evaluate the performance of the condensed graph by training a GCN on the condensed graph and testing it on the original graph. Our GCN employs a 2-layer architecture with 256 hidden units, and we set the weight decay to 1e-5 and dropout to 0.5. In the node condensation module, we utilize a 3-layer MLP with 256 hidden units and no dropout for Cora, and a 4-layer MLP with 256 hidden units and 0.5 dropout for the others. And in the link prediction model, we utilize a 3-layer MLP with 256 hidden units and 0 dropout. The learning rates for MLP parameters are both set to 0.01. During the pre-training of the link prediction model, we use a mini-batch of positive and negative edges during each epoch with the number of negative edges typically set to three times that of the positive edges. And the aggregator we utilize in this stage is a max aggregator. For the transductive datasets Cora, Ogbn-arxiv, Ogbn-products, the labeling rates are 5.2%, 53%, and 8%, respectively. We choose reduction rate $r$ {25%, 50%, 100%}, {0.1%, 0.5%, 1%}, {0.25%, 0.5%, 1%} of the label rate, with corresponding final reduction rates of being {1.3%, 2.6%, 5.2%}, {0.05%, 0.25%, 0.5%} and {0.02%, 0.04%, 0.08%}. And in the inductive datasets Reddit and Reddit2, the reduction rates are both set to {0.05%, 0.1%, 0.2%}. For DisCo, we perform 1500 epochs for Cora and Ogbn-arxiv, and 2500 epochs for the remaining datasets.

## C  Time Complexity Analysis

Assuming the number of nodes and edges in the original graph is $N$ and $E$, the node number in the condensed graph is $N'$. Assume $d$ as node feature dimensions, and there are $L$ layers with $d$ hidden units for the node condensation MLP, edge translation MLP and GNN in other methods. The number of sampled neighbors per node is $r$ in GCOND, the anchor number is $M$ in DisCo, $P$ is the number of the experts in SFGC. Considering there are $T$ iterations and $K$ different initializations in GCOND and SFGC.

The complexity of GCOND includes $TKO(LN'^2 d + LN' d)$ on the condensed graph and $TKO(r^L N d^2)$ on the original graph. The complexity of SFGC includes $TKO(LN' d^2 + LN' d)$ on the condensed graph and $TPO(r^L N d^2)$ on the original trajectories. Disco's complexity includes $TO(N' L d^2 + N + (1 + M)N')$ in the node condensation stage and $TO(ELd^2)$ in the edge translation stage. From the analysis above, it becomes evident that our method offers an improvement of at least one order of magnitude, which explains the the accelerated performance of our approach compared to gradient-matching and trajectory-matching methods.

## D  Theoretical Analysis

### D.1  Node Condensation

The initial condensed nodes are directly sampled using K-Center from the original graph, so given the K-center node features $X$, the

**Table 7: The statistics of six datasets.**

| Dataset | Task Type | #Nodes | #Edges | #Classes | #Training | #Validation | #Test |
|---|---|---|---|---|---|---|---|
| Cora | Node Classification (transductive) | 2,708 | 10,556 | 7 | 140 | 500 | 1,000 |
| Ogbn-arxiv | Node Classification (transductive) | 169,343 | 1,166,243 | 40 | 90,941 | 29,799 | 48,603 |
| Ogbn-products | Node Classification (transductive) | 2,449,029 | 61,859,140 | 47 | 196,615 | 39,323 | 2,213,091 |
| Ogbn-papers100M | Node Classification (transductive) | 111,059,956 | 1,615,685,872 | 172 | 1,207,179 | 125,265 | 214,338 |
| Reddit | Node Classification (inductive) | 232,965 | 114,615,892 | 41 | 153,431 | 23,831 | 55,703 |
| Reddit2 | Node Classification (inductive) | 232,965 | 23,213,838 | 41 | 153,932 | 23,699 | 55,334 |

condensed features $X'$ are perturbed versions of $X$, i.e.,

$$X' = X + \Delta X,$$

where $\Delta X$ represents a small perturbation constrained by anchor attachment and centroid alignment. The loss function is denoted by $\mathcal{L}(f_{\phi^*}(X), Y)$, where $f_{\phi^*}$ is a pre-trained model.

Using a first-order Taylor expansion of the loss function around $X$, we have:

$$\mathcal{L}(f_{\phi^*}(X'), Y') \approx \mathcal{L}(f_{\phi^*}(X), Y) + \nabla_X \mathcal{L}(f_{\phi^*}(X), Y) \cdot \Delta X + O(\|\Delta X\|^2).$$

Since the model $f_{\phi^*}$ was pre-trained to minimize the loss $\mathcal{L}(f_{\phi^*}(X), Y)$, we have:

$$\nabla_X \mathcal{L}(f_{\phi^*}(X), Y) \approx 0.$$

Thus, the first-order term vanishes, and we are left with:

$$\mathcal{L}(f_{\phi^*}(X'), Y') \approx \mathcal{L}(f_{\phi^*}(X), Y) + O(\|\Delta X\|^2).$$

This means the change in the loss function depends on the magnitude of $\|\Delta X\|$.

Consider the anchor number to be 1, the anchor attachment term $A(X')$ is:

$$A(X') = \|X' - X\| = \|\Delta X\| \leq \epsilon.$$

Since both anchor attachment and centroid alignment terms enforce small perturbations, $\epsilon$ is a small constant. We have:

$$O(\|\Delta X\|^2) = O(\epsilon^2),$$

$$\mathcal{L}(f_{\phi^*}(X'), Y') \approx \mathcal{L}(f_{\phi^*}(X), Y) + O(\epsilon^2).$$

As a results, the second term introduces flexibility, allowing the condensed nodes to capture more informative and representative features without deviating significantly from the basic distribution $(X, Y)$ of the first term.

## D.2 Edge Translation

The original edge existence is:

$$a_{ij} = \text{Sigmoid}\left(\frac{\text{MLP}_\vartheta(Z_1) + \text{MLP}_2(Z_2)}{2}\right),$$

where $Z_1 = [X_i; X_{N(i)}; X_j; X_{N(j)}]$ and $Z_2 = [X_j; X_{N(j)}; X_i; X_{N(i)}]$. Suppose two nodes are initially derived from $X_i$ and $X_j$. The condensed edge existence can then be expressed as

$$Z'_1 = Z_1 + \Delta Z_1 = [X_i + \Delta X_i; X_{N(i)} + \Delta X_{N(i)}; X_j + \Delta X_j; X_{N(j)} + \Delta X_{N(j)}],$$

$$Z'_2 = Z_2 + \Delta Z_2 = [X_j + \Delta X_j; X_{N(j)} + \Delta X_{N(j)}; X_i + \Delta X_i; X_{N(i)} + \Delta X_{N(i)}],$$

such that

$$a'_{ij} = \text{Sigmoid}\left(\frac{\text{MLP}_\vartheta(Z'_1) + \text{MLP}_2(Z'_2)}{2}\right).$$

Using the mean value theorem for the Sigmoid function, the difference between the original and condensed edge existence prediction $\Delta a_{ij} = a'_{ij} - a_{ij}$ is:

$$\text{Sigmoid}'(\xi) \cdot \left(\frac{\text{MLP}_\vartheta(Z'_1) + \text{MLP}_\vartheta(Z'_2)}{2} - \frac{\text{MLP}_\vartheta(Z_1) + \text{MLP}_\vartheta(Z_2)}{2}\right).$$

where $\xi$ is some value between $(Z_1, Z_2)$ and $(Z'_1, Z'_2)$.

The derivative of the Sigmoid function is:

$$\text{Sigmoid}'(x) = \text{Sigmoid}(x) \cdot (1 - \text{Sigmoid}(x)).$$

which is bounded by

$$0 \leq \text{Sigmoid}'(x) \leq \frac{1}{4}.$$

Thus, we can bound the change in $a'_{ij}$ as follows:

$$|\Delta a_{ij}| \leq \frac{1}{4} \cdot \left|\frac{\text{MLP}_\vartheta(Z'_1) + \text{MLP}_\vartheta(Z'_2) - (\text{MLP}_\vartheta(Z_1) + \text{MLP}_\vartheta(Z_2))}{2}\right|.$$

$\text{MLP}_\vartheta$ is a continuous and differentiable function. As demonstrated in the node condensation theoretical analysis, the perturbations $\Delta X_i$ and $\Delta X_j$ are small. $\Delta X_N(i)$ and $\Delta X_N(j)$ are even smaller because neighbor features are smoothed during aggregations. So we have:

$$\text{MLP}_\vartheta(Z'_1) \approx \text{MLP}_\vartheta(Z_1) + \nabla \text{MLP}_\vartheta(Z_1) \cdot \Delta Z_1$$

and

$$\text{MLP}_\vartheta(Z_2) \approx \text{MLP}_\vartheta(Z_2) + \nabla \text{MLP}_\vartheta(Z_2) \cdot \Delta Z_2.$$

Finally, the difference can be approximated as:

$$\Delta a_{ij} \approx \left|\frac{\nabla \text{MLP}_\vartheta(Z_1) \cdot \Delta Z_1 + \nabla \text{MLP}_\vartheta(Z_2) \cdot \Delta Z_2}{8}\right|.$$

The change in $\text{MLP}_\vartheta$ due to a small perturbation $\Delta Z$ is small, so as $\Delta a_{ij}$, which proves that the condensed edge existence $a'_{ij}$ remains close to the original prediction. Consequently, the connectivity pattern and topology of the original graph are effectively preserved, allowing the condensed graph to train GNNs similarly to the original one, thereby achieving good results.

## E More Experiments

## E.1 Neural Architecture Search

In this section, we aim to evaluate the effectiveness of our approach in conducting NAS. We conduct a NAS experiment on the condensed graph and compare it to a direct search on the original graph. Our investigation focuses on a search space that includes 162 GCN architectures with different configurations of layers, hidden units, and activation functions. To assess the performance of the

**Table 8: Evaluation of Neural Architecture Search. Performance is reported as test accuracy (%). "BT" and "AT" refer to evaluation before tuning and after tuning. "WD" refers to searching with the whole dataset.**

|  | GNN$_S$ | | GNN$_{NAS}$ | | Time | |
|---|---|---|---|---|---|---|
|  | BT | AT | DisCo | WD | DisCo | WD |
| Ogbn-arxiv (0.5%) | 65.6 | 66.8 | 72.1 | 72.1 | 15s | 104s |
| Reddit (0.2%) | 92.1 | 92.6 | 94.5 | 94.6 | 32s | 518s |

search process, we employ two benchmark datasets: Ogbn-arxiv and Reddit. The procedure for conducting the search on the condensed graph is outlined as follows: (1) Utilize the original graph or condensed graph to train 162 GNNs (GNN$_S$) using different GNN architectures from the search space. (2) Select the GNN architecture of the best-performing GNN$_S$ using the original validation set as the optimal GNN architecture. (3) Train the optimal GNN$_S$ architecture on the original graph and obtain the NAS result GNN$_{NAS}$.

By comparing the performance of GNN$_{NAS}$ obtained from both the condensed graph and the original graph, we can evaluate the effectiveness of the condensed graph approach in guiding the NAS process. The comprehensive search space encompasses a wide range of combinations involving layers, hidden units, and activation functions: (1) **Number of layers**: We search the number of layers in the range of {2, 3, 4}. (2) **Hidden dimension**: We search the number of hidden dimensions in the range of {128, 256, 512}. (3) **Activation function**: The available activation functions are: {Sigmoid($\cdot$), Tanh($\cdot$), Relu($\cdot$), Softplus($\cdot$), Leakyrelu($\cdot$), Elu($\cdot$)}. Table 8 summarizes our research findings, including the test accuracy of GNN$_S$ before and after tuning (we utilize the results obtained in the baseline comparison section as the pre-tuning results), the NAS result GNN$_{NAS}$, and the average time required to search a single architecture on both the condensed and original graphs.

Our experimental results demonstrate that the condensed graph of DisCo offers efficient parameter tuning and performance improvements of over 0.5% for both the Ogbn-arxiv and Reddit datasets. The best architectures discovered through NAS on the condensed graphs are {3, 512, elu($\cdot$)} and {2, 128, tanh($\cdot$)} for Ogbn-arxiv and Reddit, respectively. Remarkably, we observe that the best architectures obtained by the condensed graphs achieve comparable GNN$_{NAS}$ performance to those obtained using the original graph, with a negligible NAS result gap of less than 0.1%. Furthermore, utilizing the condensed graph significantly reduces the search time, with a speed improvement of more than six times compared to the original graph. These findings highlight the viability and efficiency of leveraging the condensed graph for NAS, as it provides comparable performance while substantially reducing the required time.

## E.2 Hyperparameter Robustness

DisCo has three hyperparameters $\alpha$, $\beta$ and $\delta$ in DisCo. Our experiments suggest these parameters are straightforward to tune. We've found effective performance across datasets by setting $\alpha = \{50, 100\}$, $\beta = \{1, 2, 5\}$, and $\delta = \{0.9, 0.95, 0.99\}$. The $\alpha$ term is crucial due to small class centroid distances, requiring significant weight ($>=50$) to preserve class distribution similarity. $\beta$ is kept small relative to

**Table 9: The performance comparison between different hyperparameters on Ogbn-arxiv (0.05%).**

| $\lambda$ | $\beta$ | $\delta$ | Acc (%) |
|---|---|---|---|
| 50 | 1 | 0.95 | 64.12 |
| 50 | 1 | 0.99 | 64.18 |
| 50 | 2 | 0.95 | 63.25 |
| 50 | 2 | 0.99 | 63.14 |
| 100 | 1 | 0.95 | 63.98 |
| 100 | 1 | 0.99 | 63.93 |
| 100 | 2 | 0.95 | 63.74 |
| 100 | 2 | 0.99 | 63.90 |
| 100 | 5 | 0.95 | 63.42 |
| 100 | 5 | 0.99 | 63.76 |

$\alpha$ ($<=0.05 \alpha$) to avoid the condensed nodes being mere samples, preserving the condensed graph's representational power. Lastly, $\delta$ performs well above 0.9. Our results indicate that setting these hyperparameters according to these guidelines consistently yields good outcomes, as shown in Table. 9.

