# OpenReview forum: "Disentangled Condensation for Large-scale Graphs"
_ACM.org/TheWebConf/2025/Conference — WWW 2025 Poster_

### Official Review · Reviewer_ShCw · 2024-11-17

**Novelty:** 3
**Technical Quality:** 5

**Review:**

## Comment

This paper proposes a novel disentangled graph condensation method that is scalable to large graph datasets. The main idea involves condensing node features and generating edges using distribution matching and link prediction modules, respectively.

### Strength:

1. Due to its strong scalability, this is the first work to conduct experiments on a large dataset (ogbn-paper100M).
2. The k-nearest pseudo-neighbor approach is a great design as it reuses the k-NN results in both the node condensation and edge translation modules.

### Weakness:

1. The novelty is limited for the following reasons:
    1. Class Centroid Alignment appears equivalent to GCDM [1] and similar to SimGC [2]. Please clarify the differences, or if there are none, acknowledge these prior works.
    2. The aggregation-and-concatenate strategy in the pretraining stage of edge translation is not novel.
2. The ablation study is missing. It would be beneficial to demonstrate the effectiveness of each module in node condensation and, most importantly, prove the necessity of the edges in condensation graph.
3. The usage of "disentangled" and "entangled" in this paper is confusing. The title's "Disentangled" supposedly means eliminating the need to train GNNs simultaneously with node condensation and edge translation (Line 272). However, this claim is unclear for two reasons. First, most graph condensation methods use MLPs, not GNNs, to generate edges. Second, many structure-free methods like GCDMX [1] and SFGC [3] don't need to generate edges at all. Moreover, in Line 99, the paper defines "entangled" as optimizing redundant parameters simultaneously. Yet, SFGC also pretrains a GCN—just like DisCo—which doesn't involve simultaneous parameter optimization.
4. The k-NN strategy heavily relies on the homophily assumption. Consequently, DisCo may not generalize well to heterophilous graphs.
5. Reddit and Reddit2 datasets are highly similar, differing only in edge numbers. Testing on both is redundant. It would be more informative to experiment with other datasets such as Pubmed, Yelp, or Amazon.
6. The source of baseline performance is unclear. Please specify how you reproduced the baselines and provide the code. If results are from original papers, indicate this clearly. For instance, some of SFGC's results appear to be copied from the original paper, while others differ.
7. It would be beneficial to include a structure-free version as an ablation study to demonstrate the effectiveness of the edge translation module.

Some Minor issues:

1. The λ should be α in Table 9.
2. It would be better to compare a sequence of similar data using a bar chart instead of Table 9.
3. Line 454: "Multilayer Perceptron" should be "Multi-Layer Perceptron" as you use MLP as an abbreviation.
4. Line 458: "binary cross-entropy" should be capitalized as you define its abbreviation.
5. Lines 1207 & 1217: MLP_2 should be MLP_θ.

[1] Liu, Mengyang, et al. "Graph condensation via receptive field distribution matching." *arXiv preprint arXiv:2206.13697* (2022).
[2] Xiao, Zhenbang, et al. "Simple graph condensation." *Joint European Conference on Machine Learning and Knowledge Discovery in Databases*. Cham: Springer Nature Switzerland, 2024.

[3] Zheng, Xin, et al. "Structure-free graph condensation: From large-scale graphs to condensed graph-free data." *Advances in Neural Information Processing Systems* 36 (2024).

**Questions:**

Please refer to the points above. I would consider raising my score if the authors could adequately address my concerns regarding Weaknesses 1-3.

**Reviewer Confidence:**

4: The reviewer is certain that the evaluation is correct and very familiar with the relevant literature

**Scope:**

4: The work is relevant to the Web and to the track, and is of broad interest to the community

---

### Official Review · Reviewer_dP2X · 2024-11-30

**Novelty:** 4
**Technical Quality:** 5

**Review:**

1. This work pinpoints the high training cost of large-scale graphs, analyzes the solution method of graph condensation as well as the scalability insufficiency problem of the current entanglement optimization paradigm, and proposes a framework of decoupled graph coalescence, which decouples the graph coalescence process into two phases of node condensation and edge translation, which is an innovative attempt in the field of graph condensation.
2. Compared with other methods, the coalescence time is shortened while ensuring the accuracy.
3. The article introduces each module is, from the goal to the realization method, logically coherent and easy for readers to understand.

**Questions:**

1. In practice, how does the specific selection and parameterization of the link prediction models used ensure consistent performance on different datasets?
2. What role does the pre-trained link prediction model play in the edge generation module? What is its specific construction and optimization process?

**Reviewer Confidence:**

3: The reviewer is confident but not certain that the evaluation is correct

**Scope:**

3: The work is somewhat relevant to the Web and to the track, and is of narrow interest to a sub-community

---

### Official Review · Reviewer_BCYU · 2024-12-01

**Novelty:** 3
**Technical Quality:** 4

**Review:**

**Pros:**
- DisCo achieves 10x faster condensation than state-of-the-art methods and handles large-scale datasets (e.g., Ogbn-papers100M) effectively.
- The proposed method is robust across multiple GNN architectures, making it versatile for various applications.
- The paper clearly defines its contributions, making it easy to understand its impact.

**Cons:**
- Although there is a certain improvement in efficiency, the overall innovation of the method is limited.
- The performance of DisCo in the overall experimental evaluation is unstable, and a clear explanation can help to understand the effectiveness of DisCo.
- While DisCo scales well, the paper does not thoroughly discuss potential trade-offs (e.g., accuracy vs. computation for extreme reduction rates).
- The author did not provide the code, which prevents me from assessing the reproducibility of the method.

**Questions:**

See cons.

**Reviewer Confidence:**

3: The reviewer is confident but not certain that the evaluation is correct

**Scope:**

3: The work is somewhat relevant to the Web and to the track, and is of narrow interest to a sub-community

---

### Official Review · Reviewer_igYF · 2024-12-03

**Novelty:** 5
**Technical Quality:** 6

**Review:**

### Summary

This paper introduces DisCo, a novel GNN-free disentangled graph condensation framework that addresses the scalability challenges of existing graph condensation methods. DisCo separates the node condensation and edge translation processes, eliminating the need for simultaneous GNN optimization and significantly reducing computational complexity. Through extensive experiments on medium to large-scale graphs, including the massive Ogbn-papers100M dataset, DisCo demonstrates superior scalability and performance, outperforming state-of-the-art methods with up to 5% higher accuracy on certain datasets. The framework's effectiveness is further validated through downstream task performance and ablation studies, highlighting its potential for practical applications in handling large-scale graph data.


### Pros

1. The methodology presented in this paper is simple, intuitive, and effective. It completely avoids the complex bi-level optimization inherent in mainstream GC methods, significantly reducing the time consumption associated with GC.
2. DisCo is highly GPU memory-efficient, whereas baseline GC methods often run out of memory with even modest reduction rates.
3. The inclusion of graph neural architecture search (NAS) experiments enhances the practical significance of DisCo.

### Cons

1. Have the authors considered further ablation studies, such as structure-free experiments (using only the condensed node features), to test the performance of DisCo?
2. In Table 1, does the author use a simple $MLP([x_i;x_j])$ link prediction model trained by BCE loss?
3. On Line 447, the authors mention using fixed aggregators like mean or sum to aggregate the features of 1-hop neighboring nodes. I am curious about two scenarios: (1) If not just 1-hop but multiple hops are aggregated, similar to Hop2Token [1], then we have $H_v=[h_v;h_{\mathcal{N}^1(v)};...;h_{\mathcal{N}^k(v)}]$. Would this bring any performance improvement? (2) Even with the above scheme, I believe it's highly lossy, as the neighbor information is simply averaged. If a GNN were used here, would there be a significant performance gain? (I understand that a major contribution of this work is being GNN-free; just out of curiosity :)
4. I'm aware that some methods, such as GEOM [2], perform better numerically than this work. This slightly reduces the contribution of the paper, though not being a dealbreaker.

Despite these queries, I believe overall they do not detract from the paper being a good work, with a simple methodology and convincing experiments.

[1] NAGphormer: A Tokenized Graph Transformer for Node Classification in Large Graphs
[2] Navigating Complexity: Toward Lossless Graph Condensation via Expanding Window Matching

**Questions:**

See cons.

**Reviewer Confidence:**

4: The reviewer is certain that the evaluation is correct and very familiar with the relevant literature

**Scope:**

4: The work is relevant to the Web and to the track, and is of broad interest to the community

---

### Official Review · Reviewer_Sn7R · 2024-12-04

**Novelty:** 5
**Technical Quality:** 6

**Review:**

The paper focuses on the graph condensation problem, and introduces a GNN-free framework DisCo that disentangles node and edge processing to improve scalability and efficiency.

Pros:
1. Comprehensive Experiments: The authors provide a complete evaluation and ablation studies for the method across diverse datasets and conditions.
2. Clarity: The paper is well-organized and easy to follow.
3. Significance: The proposed DisCo method addresses the scalability bottleneck in graph condensation. The results are impressive,


Cons:
1. It seems that the proposed DisCo approach underperforms on small datasets, such as Cora.
2. It would be better if the author could also visualize the condensed graph and check if it exhibits structure mimicking the original graph.

**Questions:**

Please refer to cons.

**Reviewer Confidence:**

3: The reviewer is confident but not certain that the evaluation is correct

**Scope:**

4: The work is relevant to the Web and to the track, and is of broad interest to the community